# Faecal metabolites as a readout of habitual diet capture dietary interactions with the gut microbiome

Robert Pope [1], Alessia Visconti [1,2], Xinyuan Zhang[1], Panayiotis Louca [1,3], Andrei-Florin Baleanu[1], Yu Lin [1], Francesco Asnicar[4], Kate Bermingham[5,6], Kari E. Wong[7], Gregory A. Michelotti [7], Jonathan Wolf [6], Nicola Segata [4], Sarah E. Berry [5,6], Tim D. Spector [1,6], Emily R. Leeming[1], Rachel Gibson[5], Cristina Menni [1,8,9,10] & Mario Falchi [1,10] ✉

The interplay between diet and gut microbiome composition is complex. Faecal metabolites, the end products of human and microbial metabolism, provide insights into these interactions. Here, we integrate faecal metabolomics, metagenomics, and habitual dietary data from 1810 individuals from the TwinsUK and 837 from the ZOE PREDICT1 cohorts. Using machine learning models, we find that faecal metabolites accurately predict reported intakes of 20 food groups (area under the curve (AUC) > 0.80 for meat, nuts and seeds, wholegrains, tea and coffee, and alcohol) and adherence to seven dietary patterns (AUC from 0.71 for the Plant-based Diet Index to 0.83 for the Dietary Approaches to Stop Hypertension score). Notably, the faecal metabolome is a stronger predictor of atherosclerotic cardiovascular disease risk (AUC = 0.86) than the Dietary Approaches to Stop Hypertension score (AUC = 0.66). We identify 414 associations between 19 food groups and 211 metabolites, that significantly correlate with microbial α-diversity and 217 species. Our findings reveal that faecal metabolites capture mediations between diet and the gut microbiome, advancing our understanding of diet-related disease risk and informing metabolite-based interventions.

Diet is well recognised for its influence on human health and gut microbiome composition[1–3]. Among dietary factors, overall dietary pattern has been shown to correlate more strongly with gut microbiome composition than individual nutrients[4–6]. Indeed, identifying specific interactions between dietary components and microbial species from the complex network of genetic, dietary, lifestyle, and environmental factors remains challenging[3]. The gut microbiome itself is closely linked to clinical markers of health, with depleted microbial diversity observed in various disease statuses[7–10]. Moreover, microbial metabolites such as short-chain fatty acids (SCFA) and secondary bile acids have been well explored in the context of health[11,12]. As the gut microbiome is modifiable through dietary interventions, it has attracted considerable interest as a therapeutic target to improve and regulate host metabolic homoeostasis[13,14].

[1]Department of Twin Research & Genetic Epidemiology, King's College London, London, UK. [2]Centre for Biostatistics, Epidemiology, and Public Health, Department of Clinical and Biological Sciences, University of Turin, Turin, Italy. [3]Human Nutrition & Exercise Research Centre, Newcastle University, Newcastle, UK. [4]Department CIBIO, University of Trento, Trento, Italy. [5]Department of Nutritional Sciences, King's College London, London, UK. [6]Zoe Limited, London, UK. [7]Metabolon, Research Triangle Park, Morrisville, NC, USA. [8]Department of Pathophysiology and Transplantation, Università Degli Studi di Milano, Milan, Italy. [9]Fondazione IRCCS Cà Granda Ospedale Maggiore Policlinico, Angelo Bianchi Bonomi Hemophilia and Thrombosis Center, Milan, Italy. [10]These authors jointly supervised this work: Cristina Menni, Mario Falchi. ✉e-mail: mario.falchi@kcl.ac.uk

Omics technologies offer new avenues to evaluate dietary exposures and study the relationship between nutritional intake and gut microbial metabolism. Among these, metabolomics is particularly powerful in providing functional insight into both microbial metabolism and dietary intake. Serum, plasma, and urinary metabolites have been extensively studied as objective measures of food consumption, adherence to different dietary patterns[15–19] and indicators of dietary related microbial metabolic pathways[20,21]. More recently, faeces has gained attention as a potential biological matrix for nutritional profiling, through the use of food-derived DNA in stool to estimate dietary intake[22]. Analysing food-derived DNA provides high-resolution data on the genetic content of foods consumed but is limited in distinguishing food items from the same source with differing nutritional profiles, such as milk and meat from the same animal. Faecal metabolomics has also shown potential for dietary exposure assessment identifying faecal biomarkers for the consumption of specific foods[23], as well as altered metabolic profiles relevant to healthy and unhealthy dietary patterns[24,25], highlighting its potential in nutritional research. We previously reported that the faecal metabolome acts as a functional readout of gut microbiome activity[26], and can mediate host-microbiome interactions[27]. By capturing both microbial metabolism and host-diet interactions, faecal metabolomics may provide a closer link to host health. Nevertheless, the broader functional associations between diet-derived faecal metabolites and their interactions with the gut microbiome remain incompletely characterised.

Dietary-related faecal metabolites, reflecting the combined metabolic activity of the gut microbiota and the host, offer a novel data source for gaining quantitative insights into the mechanisms underlying the interplay between diet and the gut microbiome. In depth exploration of these mechanisms is crucial to understand how gut microbial dynamics can be modulated either through diet or by supplementation of relevant dietary-derived metabolites[28–30]. For example, dietary fibre and its degradation by the gut microbiome has been shown to favour the microbial production of beneficial tryptophan metabolites[31]. Manipulation of the gut microbiome through dietary-derived metabolites is an appealing approach in personalised settings, particularly for individuals with specific dietary requirements or medical constraints, such as food allergies, or limited access to certain food. While it is the overall diet or whole foods that mediate the impact on gut microbial composition, targeted effects might also be achieved through selected dietary-derived metabolites that are substrates for certain microbial species, an approach that underpins prebiotic development.

Here, we used dietary data for 2647 individuals to (i) investigate how accurately the faecal metabolome predicts adherence to healthy and unhealthy dietary patterns as well as habitual consumption of food and beverage groups; (ii) identify faecal metabolite signatures of habitual food and beverage group consumption; (iii) utilise dietary associated faecal metabolites to infer the potential mediatory relationships between diet, gut microbial species and faecal metabolites. To the best of our knowledge, this is the largest study to date to explore the faecal metabolome in nutritional research, integrating dietary data with faecal metabolomics to bridge the gap between diet and the gut microbiome.

## Results
### Participants
The flowchart of the study design is depicted in Fig. 1. In total, we included 1810 participants of European ancestry from TwinsUK (mean age = 61.0 ± 13.7 years old, mean body mass index (BMI) = 25.9 ± 4.8 kg/m², 86.0% female) and 837 participants (mean age = 46.3 ± 11.6 years old, mean BMI = 25.8 ± 5.2 kg/m², 74.3% female, 91.9% European ancestry) from the ZOE PREDICT1 study. All participants had habitual dietary data, and subsets had gut metagenomics, and untargeted faecal metabolomics data (Metabolon Inc.) which included 650

metabolites, 526 of which have known chemical identity (Fig. 1). The descriptive characteristics and dietary macronutrients of the study populations are presented in Table 1.

### Faecal metabolites robustly predict diet quality and composition
To first understand how well dietary patterns are represented within the faecal metabolome, we used random forest (RF) machine learning models to predict adherence to seven a priori dietary indices. These included the Dietary Approaches to Stop Hypertension (DASH)[32], Plant-based Diet Index (PDI) and its healthful (hPDI) and unhealthful (uPDI) derivations[33], and the Alternate Mediterranean Diet Score (aMED)[34], as well as the total percentage of the diet constituted by plant- or meat-based products[4]. Two types of RF models were used: binary classifiers to predict the top and bottom quartiles of adherence to each dietary index score and regression models to predict the full range of dietary index scores. Hyperparameters, input features, and model evaluation results are detailed in Supplementary Data 1 and 2. In TwinsUK, faecal metabolite models robustly predicted adherence to dietary indices, with all indices achieving an AUC score > 0.70. Notably, the DASH diet score (AUC = 0.83 [95% CI: 0.77, 0.88], Spearman's Rho ($r_s$) = 0.52 [95% CI: 0.44, 0.59]), Total Meat% (AUC = 0.81 [0.76, 0.87], $r_s$ = 0.34 [0.25, 0.43]), and hPDI (AUC = 0.80 [0.74, 0.86], $r_s$ = 0.30 [0.20, 0.39]) showed the strongest predictive ability with AUC scores ≥ 0.80 (Fig. 2a). The AUC scores from all metabolite models were significantly higher ($p < 7.14 \times 10^{-3}$) compared to the AUC scores of their null equivalent models, which only included age, sex, and BMI as predictors (Supplementary Fig. 1). To assess how well the metabolite models generalised to new data, we tested performance using the ZOE PREDICT1 cohort. The distribution of AUC scores for the ZOE PREDICT1 cohort showed no significant differences to that achieved using the TwinsUK 20% testing set (min $p$ value from bootstrap = 0.49) (Fig. 2a, Supplementary Data 1).

Additionally, we used RF models to predict the reported habitual intakes of twenty food and beverage groups (Fig. 2a). The binary classification models effectively discriminated between high and low consumers (AUC > 0.70) of ten food groups, with the highest AUC scores achieved for reported intakes of alcoholic beverages (AUC = 0.87 [0.82, 0.92], $r_s$ = 0.53 [0.45, 0.60]), meat (AUC = 0.83 [0.77, 0.88], $r_s$ = 0.36 [0.27, 0.45]), nuts and seeds (AUC = 0.82 [0.76, 0.88], $r_s$ = 0.39 [0.30, 0.48]), tea and coffee (AUC = 0.80 [0.74, 0.86], $r_s$ = 0.42 [0.33, 0.50]) and wholegrains (AUC = 0.80 [0.74, 0.86], $r_s$ = 0.37 [0.28, 0.46]). All food and beverage group models trained in the TwinsUK dataset were tested with the ZOE PREDICT1 validation cohort, producing comparable results and thus confirming the reliability of these models (Supplementary Data 1 and 2).

### Faecal metabolites predict 10-year ASCVD risk
Among the tested indices in our study, the DASH diet score was best predicted by faecal metabolites. Given the association between the DASH diet and cardiovascular health, we sought to explore whether the faecal metabolome could further serve as a predictor for the 10-year risk of a first major atherosclerotic cardiovascular disease (ASCVD) event, as defined by the ACC/AHA guidelines[35]. A RF classification model trained using the DASH diet score and BMI showed moderate discrimination between subjects in the top and bottom quartiles of ASCVD risk (AUC DASH = 0.66 [0.59, 0.74]), and a RF regression model trained with the same predictors further showed a weak correlation between predicted risk and 10-year ASCVD risk ($r_s$ DASH = 0.22 [0.11, 0.32]). In contrast, models trained solely on faecal metabolites and BMI demonstrated superior predictive accuracy for 10-year ASCVD risk (AUC = 0.86 [0.81, 0.92], $r_s$ = 0.40 [0.30, 0.49]), significantly outperforming the model trained using the DASH diet score alone. The results of the faecal metabolite model were tested using the ZOE PREDICT1 cohort (AUC = 0.74 [CI: 0.62, 0.86], $r_s$ = 0.29

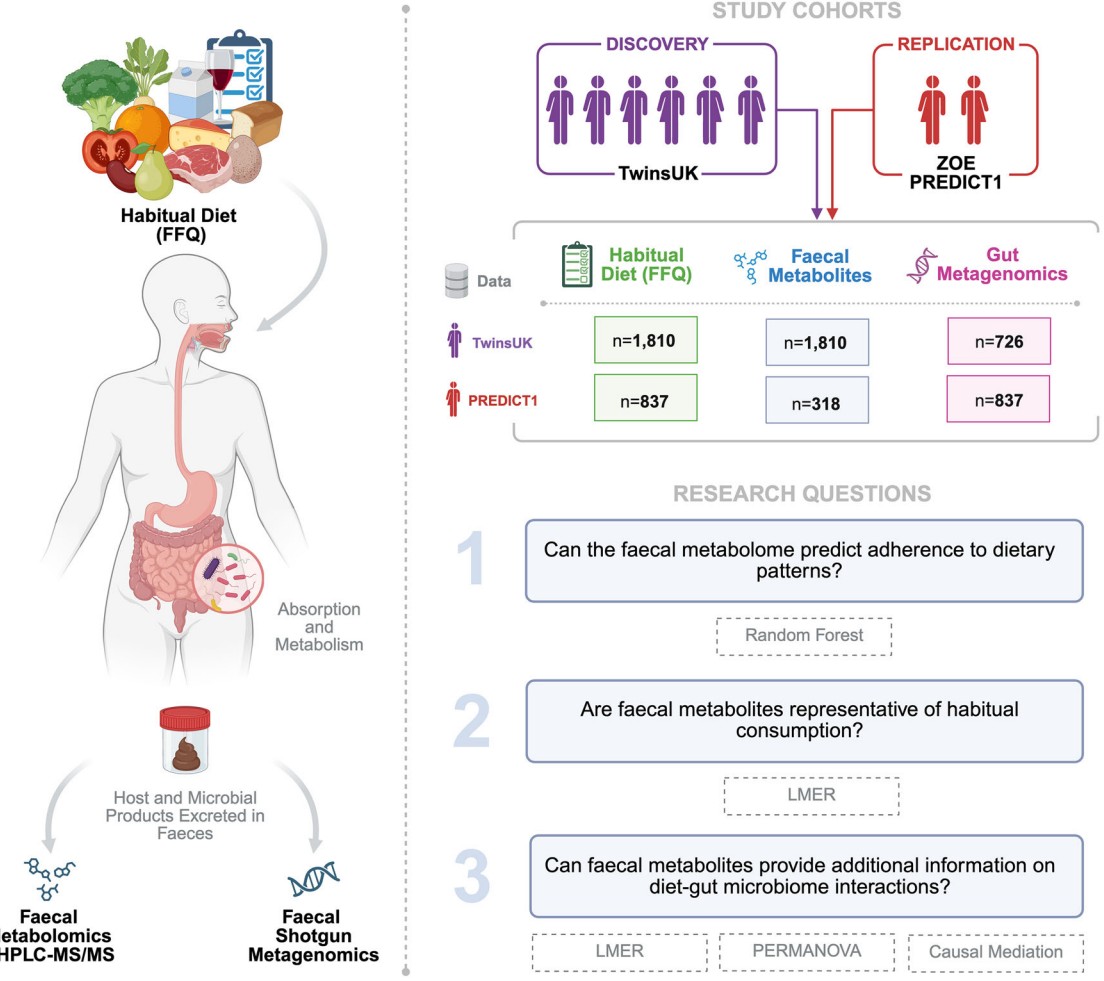

**Fig. 1 | Faecal metabolites act as a readout of habitual diet and bridge the gap between diet and the gut microbiome.** The schematic describes the discovery and replication cohorts as well as the dietary, faecal metabolomics, and metagenomics datasets, highlighting the univariate and multivariate analyses carried out in response to the aims of the study. UHPLC-MS/MS ultra high-performance liquid chromatography-tandem mass spectrometry, FFQ food frequency questionnaire, LMER linear mixed-effects regression, PERMANOVA permutational multivariate analysis of variance. Created in BioRender. Pope, R. (2025) https://BioRender.com/c51b012.

[0.10, 0.45]), showing no significant difference in AUC scores. We further tested RF models trained using faecal metabolites, BMI, and the DASH diet score (AUC = 0.86 [CI: 0.80, 0.91], $r_s$ = 0.43 [0.33, 0.51]), producing comparable results to the models trained solely with metabolites and BMI, indicating no improvement upon inclusion of the DASH diet score (Fig. 2b, Supplementary Data 1 and 2).

## A targeted panel of 54 faecal metabolites reflects dietary pattern and habitual food group consumption

Characterisation of the faecal metabolome using untargeted metabolomics is important for identifying novel findings, yet it is not always feasible or cost effective. To address this, we aimed to understand if a smaller subset of faecal metabolites could still robustly predict adherence to dietary patterns and the reported intakes of the ten food and beverage groups with AUC scores > 0.70. Using the permutation feature importance for each RF model (Supplementary Data 3), we assessed changes in predictive power through recursive feature elimination (RFE). We were able to maintain comparable AUC scores within 0.05 of the original, in both the TwinsUK testing set and ZOE PREDICT1 cohort, with significantly reduced subsets of metabolites (Supplementary Data 4). For instance, better adherence to the DASH dietary pattern was accurately predicted with just 17 metabolites, compared to

the original model which included the covariates age and BMI as well as 46 metabolites (AUC = [Original TwinsUK: 0.83, Subset TwinsUK = 0.79], Supplementary Fig. 2a). Similarly, of the ten food and beverage group models that achieved an AUC > 0.70, nine retained predictive power with smaller subsets of features (Supplementary Data 4). Specifically, alcoholic beverage consumption was well predicted with 4 metabolites (AUC = [Original TwinsUK: 0.87, Subset TwinsUK = 0.83], Supplementary Fig. 3a), with two uncharacterised metabolites as the strongest predictors (Supplementary Data 3), whilst meat intake was well predicted with just 2 metabolites (behenoylcarnitine (C22)* and arachidoylcarnitine (C20)*) (AUC = [Original TwinsUK: 0.83, Subset TwinsUK = 0.79], Supplementary Fig. 3b).

Based on the union of faecal metabolites present in each reduced subset for the dietary indices and nine food or beverage groups, we defined a panel of 54 unique metabolites that were able to maintain accurate predictions (Methods; Fig. 2c, Supplementary Data 5). This reduced metabolite panel included xenobiotic (n = 14, 25.9%), lipid (n = 13, 24.1%), uncharacterised (n = 11, 20.4%), cofactor and vitamin (n = 8, 14.8%), amino acid (n = 4, 7.4%), and energy (n = 3, 5.6%) metabolites as well as one (1.9%) partially characterised metabolite (Fig. 2d). To aid in future interpretation of the results we have included additional data on the unknown compounds in Supplementary Data 6.

**Table 1 | Descriptive characteristics and estimated daily macronutrient intakes for the study populations**

| Phenotype | TwinsUK | ZOE PREDICT1 |
|---|---|---|
| n | 1810 | 837 |
| Females, n (%) | 1556 (86.0%) | 622 (74.3%) |
| Twins present MZ: DZ: UP | 818: 520: 472 | 280: 86: 471 |
| Age (years) (SD) | 61.0 (±13.7) | 46.3 (±11.6) |
| BMI (kg/m²) (SD) | 25.9 (±4.8) | 25.8 (±5.2) |
| Weight (kg) (SD) | 69.7 (±14.4) | 73.3 (±15.6) |
| Alcohol consumers, n (%) | 1436 (79.3%) | 720 (86.0%) |
| Nutrients | | |
| Energy (kcal) | 1730.8 (±546.1) | 1,696.7 (±519.7) |
| Protein (g) | 77.6 (±25.2) | 75.3 (±23.5) |
| Total fat (g) | 65.9 (±25.8) | 70.6 (±26.1) |
| Total carbohydrate (g) | 205.0 (±73.7) | 186.8 (±69.6) |
| Non-starch polysaccharides (NSP) (g) | 17.7 (±6.6) | 16.8 (±6.2) |
| Mean alcohol (consumers) (g) | 10.9 (±12.4) | 10.2 (±10.2) |
| Median alcohol (all cohort) (g) | 5.2 (IQR: 10.6) | 6.0 (IQR: 10.0) |

Mean ± SD are reported unless indicated otherwise.

*MZ* monozygotic, *DZ* dizygotic, *UP* unpaired twin/singleton.

### The faecal metabolome is highly associated with habitual diet

Due to the observed predictive capacity of the faecal metabolome in relation to habitual diet, we explored specific faecal metabolite associations with diet in detail. We used linear mixed-effects regression models, adjusted for age, sex, BMI and accounting for twin family structure, to identify associations between individual characterised faecal metabolites and intakes of each of the twenty food and beverage groups in both cohorts. Associations with consistent effect directions across the TwinsUK and ZOE PREDICT1 cohorts were then combined using fixed-effects meta-analysis, identifying 414 significant associations between all food groups, aside from refined grains, and 211 characterised faecal metabolites ($p < 9.78 \times 10^{-5}$), (Supplementary Data 7, Supplementary Fig. 4). Of these associations, 222 were in a positive direction between 18 food groups and 143 metabolites, indicating a relationship with increased habitual intake of these food groups (Fig. 3).

### Metabolic signatures of habitual food group consumption reflect pathways relevant to their nutritional profile

We explored the significant associations identified to understand whether the observed faecal metabolites were aligned with the composition of their associated food or beverage group. The strongest associations were in a positive direction and were either relevant to the nutritional profile of the food group or the microbial byproducts of their components (Table 2). Overrepresentation analysis (ORA) of the sets of metabolites associated with each food or beverage group identified enriched metabolic pathways in line with the general nutritional profile of the food or beverage group (Table 2). These also included an enrichment for omega-3 and 6 long-chain polyunsaturated fatty acid (PUFA) metabolites with increased reported fish and seafood intake ($p = 7.02 \times 10^{-6}$).

Of the 211 dietary-associated faecal metabolites, 100 were significantly associated with multiple food and beverage groups (from 2 to 8). Hierarchical clustering of 23 faecal metabolites significantly associated with ≥4 of the food and beverage groups was sufficient to distinguish two main dietary patterns, representing divergent intakes of plant- or animal-based groups (Fig. 4a). In contrast, the remaining 111 faecal metabolites were associated with just one food or beverage group, representing unique faecal markers of habitual consumption.

(Supplementary Data 8). Of note, four caffeine metabolites were solely associated with habitual consumption of tea and coffee, faecal levels of the omega-3 PUFA metabolites, eicosapentaenoate (EPA; 20:5n3) and docosahexaenoate (DHA; 22:6n3), exclusively associated with fish and seafood consumption, and faecal creatine and creatinine were only associated with meat intake. The bacterial metabolite, urolithin A, formed from ellagitannins in nuts[36], was uniquely associated with nuts and seeds, whilst 2-aminophenol, a product of environmental microbial degradation of plant benzoxazinoids[37,38] found in wheat and rye was exclusively associated with wholegrain consumption. Finally, the alkaloid solanidine, commonly found in potatoes[39], was uniquely associated with increased intake of potato-based products and faecal saccharin, an artificial sweetener common in sweetened beverages[40], exclusively associated with the increased intake of sweetened beverages.

### Dietary pattern influences microbial species diversity

The collective impact of diet has implications for gut microbiome community structure and diversity. We evaluated the 211 dietary-associated faecal metabolites in relation to differences in microbiome diversity, first using the Shannon Index as a summary statistic of intrasample gut microbial diversity (α-diversity). In total, 55 of the 211 dietary-associated faecal metabolites were correlated with either increased or decreased gut microbiome α-diversity ($p < 2.51 \times 10^{-4}$) (Supplementary Data 9). Metabolites associated with fibre rich food groups were associated with greater α-diversity, including the dicarboxylic fatty acid (DCFA) octadecanedioate (C18-DC) ($p = 1.64 \times 10^{-21}$), the monohydroxy fatty acid 3-hydroxyoleate* ($p = 5.28 \times 10^{-11}$), and the microbial derived enterolactone[41] ($p = 1.77 \times 10^{-8}$) (Fig. 4b). In contrast, faecal levels of secondary bile acids and cholesterol metabolites, associated with increased meat consumption and decreased intakes of plant-based foods, were associated with reduced gut microbiome α-diversity (Fig. 4b).

Next, to assess how inter-individual differences in dietary associated faecal metabolite profiles correspond to variation in gut microbial community structure (β-diversity, measured using Bray-Curtis dissimilarities), we applied permutational analysis of variance (PERMANOVA). Of the dietary associated faecal metabolites, 113 metabolites for TwinsUK ($p < 2.51 \times 10^{-4}$), and 100 metabolites for ZOE PREDICT1 ($p < 2.53 \times 10^{-4}$) explained a significant amount of variance between individuals. There were 87 common metabolites across both cohorts explaining 24.9% of the variance between individuals (Supplementary Data 10). These included plant-based food metabolites that also associated with increased α-diversity, such as octadecanedioate (C18-DC), (Weighted $R^2 = 1.75\%$), 3-hydroxyoleate* (Weighted $R^2 = 1.11\%$), enterolactone (Weighted $R^2 = 0.92\%$) and additionally, 2,6-dihydroxybenzoic acid (Weighted $R^2 = 1.26\%$). The secondary bile acid and sterol metabolites that were associated with decreased α-diversity, also explained a significant proportion of the variance in β-diversity, most notably ursocholate (Weighted $R^2 = 1.72\%$), isoursodeoxycholate (Weighted $R^2 = 1.58\%$) and 25-hydroxycholesterol sulphate (Weighted $R^2 = 1.38\%$).

### Mediation analysis prioritises potential dietary mediators of the gut environment

From fixed-effects meta-analysis of results from linear mixed-effects regression models in TwinsUK and ZOE PREDICT1, we identified 205 associations between 19 food and beverage groups and 144 microbial species (FDR < 0.1) (Supplementary Data 11). In comparison, we observed 4178 significant associations between 319 microbial species and 209 dietary-associated faecal metabolites at a stricter FDR threshold (FDR < 0.01). As expected, there were markedly more associations between dietary-associated faecal metabolites and the microbiota, that were in turn stronger, than those observed directly with dietary data and microbial species, highlighting the superior

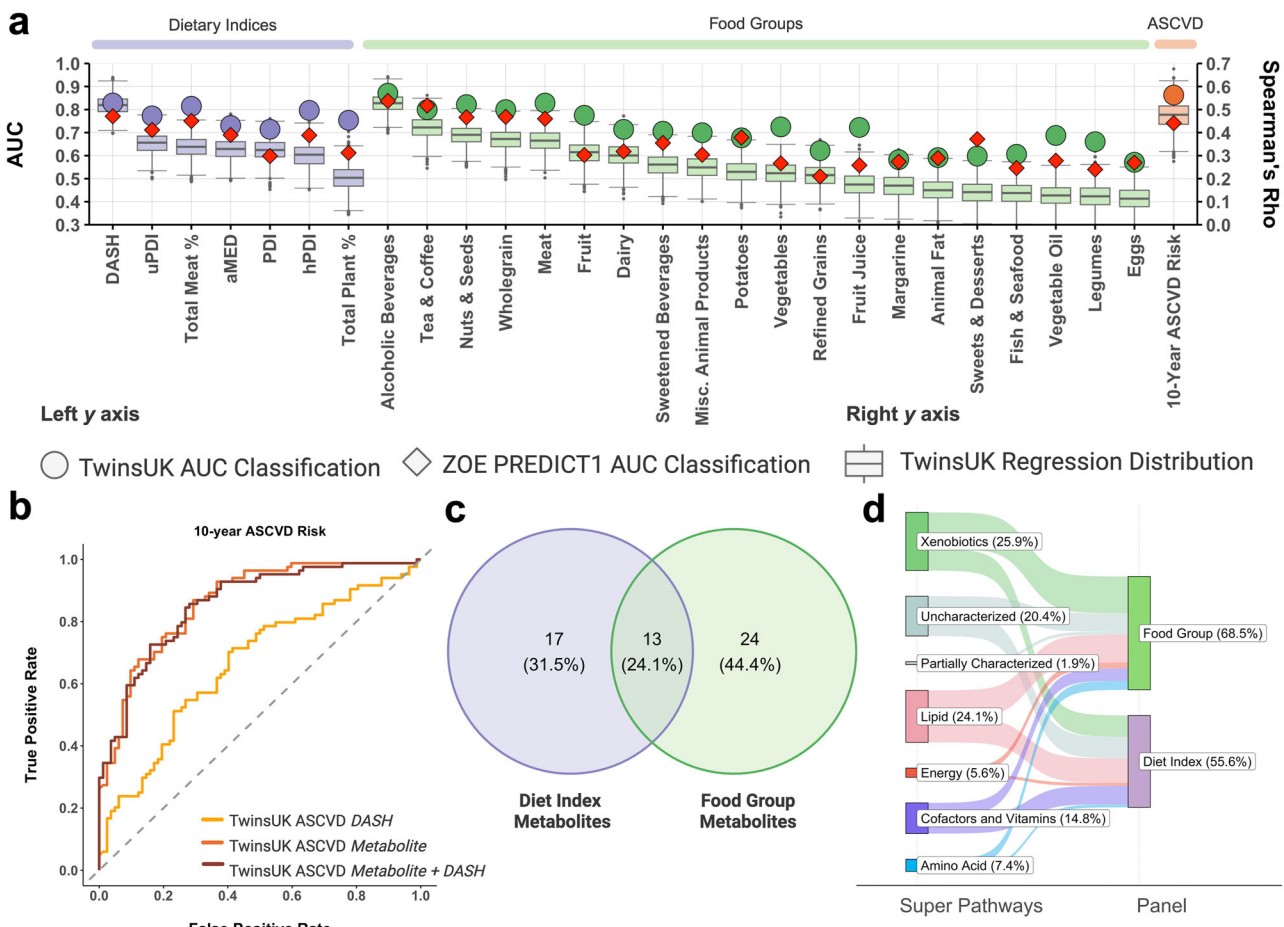

**Fig. 2 | Random forest machine learning models trained on faecal metabolite profiles accurately predict diet quality, identifying a targeted panel of 54 dietary-predictive metabolites. a** Prediction of adherence to seven dietary patterns, habitual intakes of 20 food and beverage groups, and 10-year atherosclerotic cardiovascular disease (ASCVD) risk using faecal metabolites as predictors for RF binary classification (TwinsUK $n = 905$; ZOE PREDICT1 $n = 159$) and regression models (TwinsUK $n = 1810$). The left y-axis reports the area under the curve (AUC) values for RF binary classification models (TwinsUK cohort 20% as circles; ZOE PREDICT1 as diamonds). The right y-axis reports the distribution of the Spearman's rank correlation coefficients between predicted and observed labels for the TwinsUK 20% testing set, computed by 1000 bootstrapped samples with replacement. Boxplots show the median (centre line) and interquartile range (box limits), with whiskers extending to 1.5 times the interquartile range. **b**, Receiver operating characteristic (ROC) curves for prediction of top and bottom quartiles of 10-year ASCVD risk using faecal metabolites, BMI, and the Dietary Approaches to Stop Hypertension (DASH) diet score. **c** Venn diagram of the 54 faecal metabolites identified as part of the metabolite panels for prediction of adherence to dietary patterns and habitual intake of nine food and beverage groups with an original AUC > 0.70. **d** Sankey plot of the corresponding metabolic pathways of the 54 faecal metabolites included in the panel. Created in BioRender. Pope, R. (2025) https://BioRender.com/9z4te71.

statistical power. Below the Bonferroni threshold ($p < 6.67 \times 10^{-5}$), 2620 of these associations were significant, including established correlations between microbial metabolites such as SCFAs and secondary bile acids, with their corresponding bacterial producers (Supplementary Data 12).

We observed 642 trios of mutual associations between metabolite, microbe and food or beverage group, allowing us to explore mediatory relationships in the TwinsUK cohort. We applied mediation analysis as a high-throughput screen to identify diet-derived metabolites whose statistical pattern is compatible with mediation by the gut microbiome, or as mediators themselves of the effect of diet on microbial composition (Supplementary Data 13). Among these results, two species of *Dorea* were significant mediators of faecal levels of the secondary bile acids, lithocholate (*Dorea formicigenerans* proportion of effect: 15.62%) and isohyodeoxycholate (isoHDCA) (*Dorea sp. AF36 15AT* proportion of effect: 15.41%) in response to meat consumption (Fig. 4c). Further to this the meat associated faecal metabolite, cholesterol sulphate, was suggested as a mediator on the increased abundance of *Ruminococcus torques* in response to increased meat

intake (Proportion of effect: 12.31%) (Fig. 4d). The metabolite *myo*-inositol was highlighted either as a potential mediator on the abundance of *Faecalibacterium SGB15346* (Proportion of effect: 11.11%) (Fig. 4d) in response to increased intake of nuts and seeds.

### Faecal metabolomics bridge the gap between diet and the gut microbiome

Given the extensive associations observed between the faecal metabolome and habitual diet, and in turn between the dietary-associated metabolites and gut microbial species, we aimed to understand if faecal metabolites, a quantitatively superior data source, could reveal dietary interactions with the gut microbiome that may not be captured by dietary recall data alone. Among the associated metabolites, some have been previously identified as markers of food or beverage group consumption in alternative biofluids, such as urine, serum or plasma. For instance, we validated in faeces a number of metabolites previously identified in serum as markers of habitual coffee consumption. These included the caffeine metabolites, 1,3-dimethylurate and 5-acetylamino-6-amino-3-methyluracil[42–44], as well as quinate, a

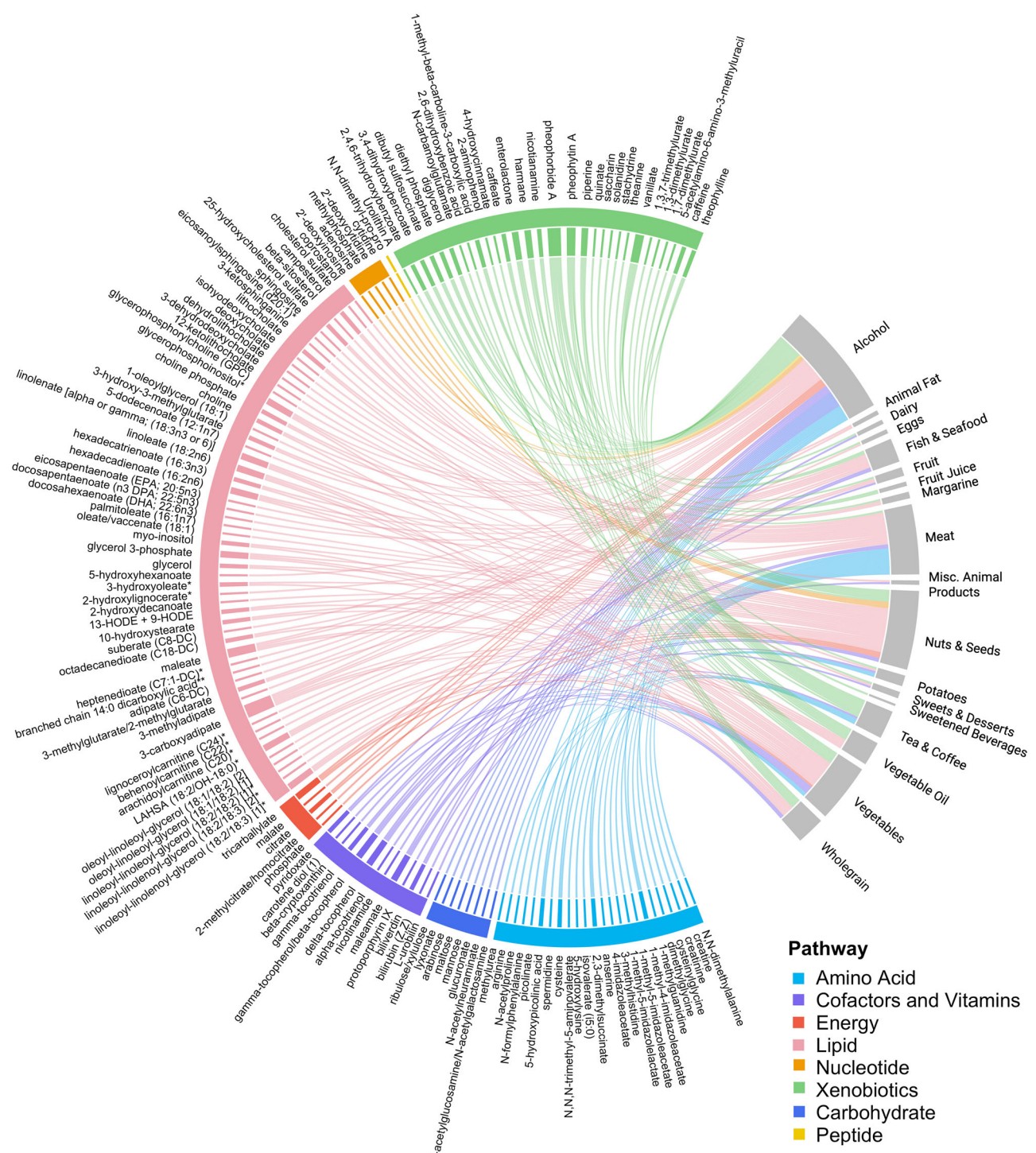

**Fig. 3 | Faecal metabolites represent a readout of habitual consumption of food and beverage groups.** Chord diagram of 222 significant positive associations between characterised faecal metabolites and food and beverage groups. Associations identified from fixed-effects meta-analysis of linear mixed-effects regression models, corrected for age, sex, BMI, and twin family structure in the TwinsUK ($n = 1810$) and ZOE PREDICT1 ($n = 318$) cohorts, were considered significant below a Bonferroni-derived threshold of $9.78 \times 10^{-5}$. Faecal metabolites are organised and coloured according to their metabolic superpathways. To facilitate the interpretation of the results we show here the positive associations, as they reflect intake of the food and beverage groups instead of dietary choices between food groups. A figure with the full set of 414 associations is depicted in Supplementary Fig. 4. Created in BioRender. Pope, R. (2025) https://BioRender.com/06wl9fo.

secondary metabolite of the coffee-derived polyphenol, quinic acid[45–47]. These faecal metabolites were strongly associated with increased abundance of *Lawsonibacter asaccharolyticus* (labelled as *Clostridium phoceensis* with the 2021 version of MetaPhlAn4) and *Massilioclostridium coli*, the two microbes showing the strongest

associations with coffee consumption in a large-scale study of 22,867[46]. However, in our study, only *Lawsonibacter asaccharolyticus* showed significant association with tea and coffee consumption while the association with *Massilioclostridium coli* would not have been identified without the faecal metabolomics data.

**Table 2 | Food and beverage groups associated with more than 20 faecal metabolites**

| Food Group | Number of associations | Pathway Enrichment (p value) | Strongest Association | Super Pathway; Sub Pathway | p value; β [95%CI] | Metabolite Description |
|---|---|---|---|---|---|---|
| Nuts and Seeds | 93 | Diacylglycerol (2.53 × 10⁻⁴) | 10-hydroxystearate | Lipid; Fatty Acid, Monohydroxy | $6.40 \times 10^{-27}$; 0.23 [0.19, 0.27] | Produced by lactic acid bacteria from the PUFA, linoleic acid, of which nuts are a dietary source[113–115]. |
| Alcoholic Beverages | 57 | - | Vanillate | Xenobiotics; Food Component/Plant | $1.22 \times 10^{-42}$; 0.28 [0.24, 0.32] | Key flavour component of alcoholic beverages[116]. |
| Meat | 41 | Secondary Bile Acid Metabolism (2.25 × 10⁻⁵) | Behenoylcarnitine (C22)* | Lipid; Fatty Acid Metabolism (Acyl Carnitine, Long Chain Saturated) | $9.26 \times 10^{-37}$; 0.25 [0.21, 0.29] | Acylcarnitine metabolites in plasma and urine have been associated with meat intake[117,118]. |
| Vegetables | 39 | Long-Chain Polyunsaturated Fatty Acid (n3 and n6) (3.37 × 10⁻⁴) | Pheophorbide A | Xenobiotics; Food Component/Plant | $6.37 \times 10^{-15}$; 0.16 [0.12, 0.20] | A chlorophyll degradation product, particularly relevant to green vegetables[119]. |
| Wholegrains | 38 | – | α-tocotrienol | Cofactors and Vitamins; Tocopherol Metabolism | $1.39 \times 10^{-20}$; 0.20 [0.15, 0.24] | A vitamin E analogue, found in grains[120]. |
| Fruit | 30 | – | β-cryptoxanthin | Cofactors and Vitamins; Vitamin A Metabolism | $4.81 \times 10^{-22}$; 0.21 [0.17, 0.25] | A pro-vitamin A carotenoid, of which citrus fruits are an important dietary source[21]. |
| Tea and Coffee | 23 | Xanthine Metabolism (1.64 × 10⁻⁹) | 5-acetylamino-6-amino-3-methyluracil | Xenobiotics; Xanthine Metabolism | $9.01 \times 10^{-36}$; 0.25 [0.21, 0.29] | A caffeine metabolite also associated with coffee intake in urine[42,122]. |

Associations derived from fixed-effects meta-analysis (two-sided) of linear mixed-effects regression models in the TwinsUK (n = 1810) and ZOE PREDICT1 (n = 318) cohorts, adjusting for age, sex, BMI, and twin family structure. Significance was assessed by means of the Bonferroni threshold derived in the TwinsUK cohort ($p < 9.78 \times 10^{-5}$). For each food or beverage group, the faecal metabolite with the strongest association is displayed, along with a description of the metabolite. Reported statistics include the meta-analysis-derived p value, the pooled β estimate and 95% confidence intervals. Metabolic pathway enrichment for the sets of significantly associated metabolites was performed using overrepresentation analysis (hypergeometric test) with the background set of all measured metabolites. Pathways with p values below the Bonferroni threshold ($p < 2.85 \times 10^{-5}$) were considered significantly enriched.

Similarly, we validated in faeces the previously reported association between wholegrain consumption and the metabolites 2,6-dihydroxybenzoic acid and 2-aminophenol, which had been reported in urine[48,49] and plasma[50,51]. In our study, these two metabolites were associated with a greater abundance of *Faecalibacterium prausnitzi*, a known fibre-degrading microbial species[52] that has been correlated with fibre-rich dietary patterns and increased wholegrain consumption[48,53,54]. Again, this association between *Faecalibacterium prausnitzi* and wholegrain intake would have not been detected from the dietary recall data alone.

Finally, to further evaluate the relevance of faecal metabolites in capturing diet–microbiome interactions, we took advantage of a subset of 1618 individuals from TwinsUK with concurrent faecal and serum metabolomics data. Although the serum metabolome was similarly as predictive as the faecal metabolome for the dietary indices and the nine food groups (Supplementary Data 14), only 2.8% of dietary-associated serum metabolites were associated with 1.2% of microbial species. In contrast, 97% of dietary-associated faecal metabolites were correlated with 58% of the identified gut microbial species, showing significantly stronger associations compared to serum metabolites (Wilcoxon rank-sum test; $p < 2.22 \times 10^{-16}$) (Supplementary Fig. 5).

## Discussion

In this system-level study, which to our knowledge is the largest to date exploring the faecal metabolome in nutritional research, we demonstrate that faecal metabolites are a powerful tool to investigate the complex relationship between diet and the gut microbiome. We show that faecal metabolic profiles reflect adherence to dietary patterns, revealing biologically plausible markers of habitual consumption and replicating established dietary biomarkers observed in serum and urine. Building on our previous studies on the interplay between faecal and serum metabolites and the gut microbiota[26,27], we integrated dietary data and faecal metabolomics to bridge the gap between diet and the gut microbiome.

Habitual dietary patterns represent the varying, and often opposing, intakes of different foods. At a granular level, we observed associations between habitual consumption of food and beverage groups and faecal metabolites that aligned with the nutritional composition of the food source. For example, regular consumption of meat was associated with metabolites of essential amino acids, cholesterol and acyl carnitine fatty acids, the latter previously shown to be elevated in the faeces of individuals following a Western diet versus a plant-based diet[55,56]. Metabolites associated with multiple food and beverage groups captured divergent intakes of plant- and animal-based foods, reflecting patterns well described in the UK population[57]. Studies, including ZOE PREDICT1, have described the same dietary patterns reflected in the composition of the gut microbiome[4,6], indicating strong parallels between dietary interactions with both the gut microbiome and the faecal metabolome.

By identifying the faecal metabolites that were most important for the RF models to predict dietary patterns and the intakes of nine food and beverage groups, we developed a targeted panel of 54 metabolites. The panel contains both characterised and uncharacterised faecal metabolites, highlighting that while relevant dietary information can be gained from known metabolites, the unmapped landscape of the faecal metabolome also holds important implications for nutritional exposure assessment. The extent of the chemical diversity of our diet remains largely uncharted[58] and future identification and quantification of unknown dietary compounds may identify additional metabolites that are relevant to the gut microbiome and human health.

Of the known metabolites identified in our study, several dietary-associated faecal metabolites correspond to biomarkers for dietary exposures in serum, plasma, or urine. Many of these faecal metabolites were also associated with specific gut microbial species, revealing

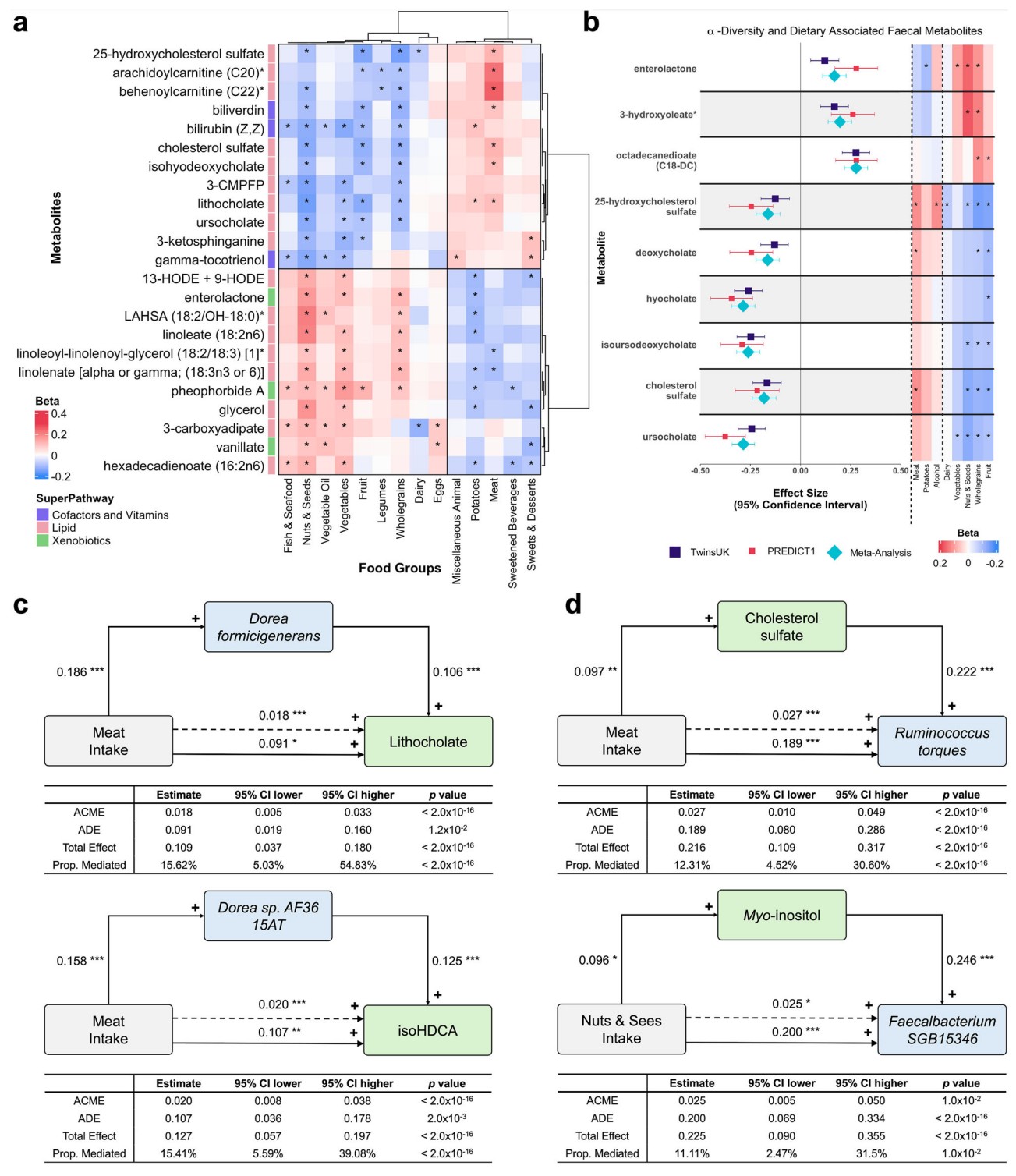

potential mechanistic links. For example, *Dorea sp. AF36-15T* mediated faecal levels of the secondary bile acid isoHDCA in response to meat intake. The secondary bile acid, isoHDCA, is formed from primary bile acids by 7α-dehydroxylating bacteria[59], such as *Dorea sp. AF36-15T*[60]. We also identified cholesterol sulphate as a mediator of *Ruminococcus torques* abundance, a mucin-degrading anaerobe associated with meat consumption[6] and shown to mediate secondary bile acids[61–64]. On the other hand, *myo*-inositol, which has been explored due to its metabolic and anti-inflammatory properties[65–67], has been implicated as a substrate for SCFA production by gut microbial species such as species from the *Anaerostipes* genus[68]. Our results show that increased

excretion of *myo*-inositol was associated with regular consumption of nuts and seeds, and a greater abundance of *Faecalibacterium SGB15346*. Nuts contain phytate, a six-fold dihydrogenphosphate ester which can be degraded by commensal gut microbes to form *myo*-insoitol[69,70]. Whilst certain *Faecalibacterium* species are known SCFA producers[71], it is unclear if *SGB15346* can ferment *myo*-inositol to produce SCFAs. Our mediation analysis suggests *myo*-inositol may influence *Faecalibacterium SGB15346* abundance, though the reverse may also be true in which the species mediates the release of *myo*-inositol upon phytate degradation. These mediation analyses cannot establish causality, and longitudinal or intervention studies are

**Fig. 4 | Dietary-associated faecal metabolites are strongly associated with gut microbiome composition and species, providing insights into diet-microbiome interactions. a** Clustered heatmap of 23 faecal metabolites (y-axis) significantly associated with ≥4 food and beverage groups for the PDI (x-axis). Associations were tested using linear mixed-effects regression models (two-sided) in TwinsUK ($n = 1810$) and ZOE PREDICT1 ($n = 318$), adjusted for age, sex, BMI, and twin family structure. Results from both cohorts were combined using fixed-effects meta-analysis. Significance was defined as meta-analysis-derived $p$ values below the Bonferroni threshold ($p < 9.78 \times 10^{-5}$). The tile colour represents the direction of association (blue = negative, red = positive); * denotes significance. Hierarchical clustering used the ACC distance metric and defined clusters based on the optimal silhouette score. Exact regression coefficients (β), standard errors, and $p$ values are reported in Supplementary Data 7. **b** Associations of faecal metabolites with gut microbiome α-diversity (forest plot) and food groups (heatmap) from fixed-effects meta-analysis of β estimates (TwinsUK = purple squares, ZOE PREDICT 1 = red squares, meta-analysis = blue diamonds). The whiskers represent the 95%

confidence intervals around the β estimates. Associations were considered significant if the meta-analysis-derived $p$ value was below the Bonferroni thresholds of $p < 2.51 \times 10^{-4}$ (α-diversity) and $p < 9.78 \times 10^{-5}$ (food groups). Exact regression coefficients (β), standard errors, and $p$ values are reported in Supplementary Data 7 and 9. Mediation analysis in the TwinsUK cohort ($n = 474$) showing direct and indirect effects of microbial species (**c**) or faecal metabolite (**d**) mediators in response to habitual diet. Linear mixed-effects regression models (two-sided) adjusted for age, sex, BMI, and accounting for twin family structure, were used to define direct (solid arrows) and indirect (dotted arrows) paths. Significance: *$p < 0.05$, **$p < 0.01$, ***$p < 0.001$. The average causal effect (ACME) and average direct effect (ADE) are summed to give the total effect with Prop. Mediated showing the percentage of the effect attributed to the mediator. The metabolite iso-hyodeoxycholate is abbreviated to isoHDCA. Test statistics are reported in Supplementary Data 13. Created in BioRender. Pope, R. (2025) https://BioRender.com/ksv93ie.

needed. Nonetheless, these findings can guide future intervention studies to identify food-derived metabolites acting as prebiotics that can modulate gut microbiome composition and function, and probiotics to modulate the production of microbial metabolites in response to food intake.

While FFQs are commonly used to estimate habitual diet and rank individuals by intake, they are inherently limited by recall bias and their semi-quantitative nature[72,73]. In contrast, food diaries and 24-h recalls can capture short-term intake in greater detail but require multiple time points to reliably reflect diet–microbiome relationships[72,74]. Notably, dietary records from several days before the stool sample show stronger correlations with microbiome composition and the food-derived DNA present in stool, compared to dietary records from the day prior[5,22]. This adds complexity to both dietary assessment and the interpretation of food–microbiome relationships, given individual variation in digestion, absorption, and transit time. Our findings show that faecal metabolites reflect habitual diet, and that diet-associated metabolites in faeces are more strongly linked to the gut microbiome than those in serum. Although metabolic markers of diet have been identified in the blood metabolome[75,76], our results indicate that faecal metabolites more directly reflect the diet–microbiome interface, offering a microbiota-proximal and functionally informative readout of dietary exposure. For instance, although both *Lawsonibacter asaccharolyticus* and *Massilioclostridium coli* have been linked to coffee intake in large cohorts[46], only *Lawsonibacter asaccharolyticus* was associated with FFQ-derived data in our study, yet both were robustly associated with coffee-derived faecal metabolites. Similarly, faecal markers of wholegrain intake revealed associations with *Faecalibacterium prausnitzii* that were not captured by FFQ data. These examples illustrate the greater sensitivity of faecal metabolite profiling in detecting biologically meaningful diet–microbiome interactions, supporting its value as a complementary tool to traditional dietary assessment methods.

Finally, by effectively capturing dietary exposure and gut microbial activity, faecal metabolite profiles more accurately distinguished individuals at high and low 10-year ASCVD risk compared with the DASH diet score, which itself is associated with improved cardiometabolic health[77,78]. This suggests that faecal metabolites capture more detailed and health-relevant information about dietary patterns than can be obtained from dietary indices alone. Both blood and faecal metabolites have been associated with health outcomes[79,80], with blood metabolites often studied in relation to dietary influences on health[81,82]. However, they reflect distinct biological processes, and interpretation requires appropriate context. Serum metabolites are shaped by tightly regulated host metabolism, whereas faecal metabolites represent the functional interface between diet and the gut microbiome. Several studies have highlighted the limited correlation

between metabolites in faeces and blood[83,84], with associations between the same metabolite and health differing depending on the biofluid[80]. Our findings support the value of faecal metabolomics in capturing both dietary patterns and gut-associated metabolic processes that contribute to disease risk, as previously shown for gastrointestinal conditions, impaired fasting glucose (within TwinsUK)[85–88], and in the present study for 10-year ASCVD risk.

The large, accurately phenotyped discovery cohort is a key strength of the study and results were replicated in an independent cohort, further strengthening our findings, however, there are some limitations. Both the UK-based discovery and replication cohorts were predominantly female and of European ancestry. This is disproportionate compared with the general UK population in which females represent 51.0% and individuals of European ancestry 83.6%[89]. Dietary assessment using FFQs has inherent limitations, such as recall bias and imprecise portion size estimation. Moreover, relying on FFQ data likely limits our ability to detect associations with metabolites influenced by recent or short-term dietary exposures. Although 24-h recalls and food diaries can offer more detailed, time-specific information, they also present challenges, including higher participant burden, variability in reporting, and the need for repeated measures to accurately reflect habitual intake[72]. In large-scale studies like ours, FFQs represent a practical tool to capture dietary patterns, which in turn have been shown to be reproducible for the long-term assessment of broad food groups and dietary patterns in large cohorts[90,91], as well as relevant to the gut environment[4,6,22]. The associations we observed between habitual diet and faecal metabolites support the notion that these long-term patterns are biologically meaningful. Future studies incorporating more precise dietary assessments or controlled feeding interventions will be valuable for validating faecal biomarkers of diet and assessing their reproducibility across different populations. The metabolism and absorption of dietary components is not uniform between individuals and food transit time was not accounted for. This also relates to the cross-sectional nature of the study, preventing the exploration of the temporal stability of faecal metabolite profiles, and highlighting an important area for future research.

In conclusion, we have integrated dietary, faecal metabolomics and gut metagenomics data from two UK-based population cohorts, demonstrating the utility of faecal metabolomics for the exploration of habitual diet and the gut microbiome. We identified faecal metabolic signatures relevant to health-related dietary patterns, as well as describing gut microbial species that mediated, or were mediated by, faecal metabolites in response to diet. Furthermore, we demonstrate the power of faecal metabolites to uncover dietary interactions with the gut microbiome that may not be captured through dietary recall methods. Our findings represent a resource for the further exploration of the faecal metabolome as an important dietary data source that can

be used to disentangle complex interactions between diet and the gut microbiome. We have further created an online database of our results for researchers to explore and facilitate future research (Data availability). These insights can inform dietary recommendations based on an individual's existing gut microbiome or alternatively used to design targeted nutritional interventions and prebiotic metabolites to modulate gut microbiome composition, ultimately improving health.

## Methods

### Study design

The study design and conduct complied with all relevant regulations regarding the use of human study participants. For TwinsUK, the volunteers provided informed consent, and the study was approved by the North West−Liverpool Central Research Ethics Committee (REC Ref19:/NW/0187) via the Integrated Research Application System (IRAS 258513). For the ZOE PREDICT1 study, ethical approval was obtained in the UK from the Research Ethics Committee (REC Ref18:/LO/0663) via the Integrated Research Application System (IRAS 236407). All individuals provided informed consent, and the trial was registered on ClinicalTrials.gov (registration number: NCT03479866). The protocol for this non-randomised, single-arm, single-blind trial, conducted between 5 June 2018 and 8 May 2019, is available on Protocol Exchange (Research Square)[92]. The trial was conducted in accordance with the Declaration of Helsinki and Good Clinical Practice. Reported sex of participants refers to sex assigned at birth.

### Discovery cohort

Study participants were individuals enroled in the TwinsUK registry, a national register of adult twins recruited as volunteers without selecting for any particular disease or trait[93]. Here we included 1810 individuals with faecal metabolomics profiling (Metabolon Inc.) who completed an adapted 131-item European Prospective Investigation into Cancer and Nutrition (EPIC) Food Frequency Questionnaire (FFQ)[73] within 3 years (mean time = 1.4 years (±1)) of their faecal sample collection.

### Replication cohort

The replication cohort was an independent sample of 837 individuals from the ZOE Personalised Responses to Dietary Composition Trial (PREDICT1) study[94]. Baseline stool samples were collected as previously described[94] and all participants completed the same adapted 131-item FFQ as the discovery cohort. Gut microbiome compositional data were generated from faecal shotgun metagenomics and a subgroup of 318 participants also had faecal metabolites measured by Metabolon Inc. The faecal metabolomic data used in this manuscript was an exploratory, non-pre-specified outcome.

The data processing pipeline detailed below for diet, metabolite, and metagenome data was applied independently to both the discovery and replication cohorts.

### Food frequency questionnaire collection and processing

Habitual dietary intake was measured using an adapted 131-item EPIC FFQ in a subset of TwinsUK participants between 2014 and 2022. The FFQ responses were processed using the FETA software (v. 2.53)[95] to determine the macro- and micronutrient data as well as daily intakes (g) for each line item. Following the framework defined by Mulligan et al.[95], FFQ records were excluded if more than 10 line items were left unanswered. FFQs were further excluded if the ratio of estimated total energy intake (kcal) to the participant's basal metabolic rate (determined by the Harris-Benedict equation[96]) was more than 2 standard deviations from the mean. Line items were aggregated into 20 food and beverage groups based on previously defined groupings for the same adapted 131-item FFQ[4] (Supplementary Table 1). To control for differences in energy intake nutrient and food group intakes were adjusted using the residual method[97].

### Dietary index calculations

Seven a priori dietary indices were calculated using the FFQ data to independently evaluate relative adherence to dietary patterns within the discovery and replication cohorts. These indices included the PDI, and its hPDI and uPDI derivations[33], the DASH[32], the aMED[34] and the total percentage of the diet constituted by plant-based foods[4] or meat products. FFQ food and nutrient categories for each index are detailed in Supplementary Tables 1–4.

### Calculation of 10-year risk of ASCVD

A pooled cohort-equations approach was used to compute an individual's 10-year risk of a first hard ASCVD event according to ACC/AHA guidelines[35]. Sex and race specific coefficients were used to multiply the following variables: age (years), total cholesterol (mg/dL), HDL cholesterol (mg/dL), mean systolic blood pressure from three consecutive measurements (mmHg), hypertension medication (Y/N), diabetes status (Y/N) and current smoking status (Y/N) (Supplementary Table 5)[35]. Systolic blood pressure and serum lipids were measured by a trained nurse and self-reported data on hypertension medication, diabetes and smoking status were collected via questionnaires[93]. The 10-year ASCVD risk score was calculated for participants from the TwinsUK and ZOE PREDICT1 cohorts with faecal metabolomics data, who were younger than 80 years old and had all relevant measurements taken within 3 years of their faecal sample (TwinsUK: $n = 1720$, ZOE PREDICT1: $n = 110$).

### Faecal sample collection

Participants from the TwinsUK cohort collected their stool samples at home. Samples were then either refrigerated for up to 2 days prior to the twins' annual clinic visit at King's College London or sent as soon as possible via post using blue Royal Mail Safe Boxes to ensure the samples were transported under the correct cooled temperature. Stool samples were stored at −80 °C at St Thomas' Hospital, London. Stool sample collection for the ZOE PREDICT1 cohort has previously been described[94]. Of the 1810 stool samples from random participants included in the study, 58 were returned to St Thomas' Hospital via post using blue Royal Mail Safe Boxes. A principal component analysis using faecal metabolite data showed the samples returned by post were indistinguishable from the samples that were returned during a TwinsUK clinic visit (Supplementary Fig. 6).

Of the study participants included with gut metagenomics and faecal metabolomics ($n = 726$), 668 (92%) of the study participants had gut metagenomics and faecal metabolomics characterised on the same faecal samples. For the remaining 58 participants (8%) the faecal samples were collected, on average, within 3.2 months. The metagenomic data included in this study were a subset of a larger dataset from TwinsUK, in which metagenomics sequencing was performed on 2075 randomly selected samples from 1679 participants collected between January 2012 and June 2019.

### Serum sample collection

Of the 1810 participants included from TwinsUK, 1618 (89.4%) had available serum metabolomics data from serum samples collected at the same time as their faecal samples. Serum samples were collected onsite as part of the routine TwinsUK participant clinic visit and stored at −80 °C at St Thomas' Hospital, London.

### DNA extraction and sequencing

The methodology used for DNA extraction, library preparation, and sequencing methods have been previously described for TwinsUK[27] and the ZOE PREDICT1[4] cohorts. For TwinsUK, in brief, a MagMax Core Lysis Solution and beating beads were added to 1 g of stool sample and vortexed for homogenisation. Proteins were degraded by a binding solution and DNA extracted by KingFisher Flex robot. DNA was washed in 2 steps by washing solutions and eluted in MagMax Core Elution

Buffer. DNA samples were quantified by Qubit, and sequencing was carried out by GenomeScan (https://genomescan.nl/) using an Illumina HiSeq 2500 with a read length of 2 × 125 bp.

## Taxonomic profiling

For TwinsUK, paired-end reads quality control was performed using the YAMP pipeline[98] (v.0.9.5.0, mode "qc") with default parameters. Samples yielding fewer than 10 million reads were excluded from further analysis. Taxonomic profiling and the quantification of organism relative abundances were performed with MetaPhlAn (v.4.beta2)[99], with the January 2021 species-specific database, which consisted of 26,970 species-level genome bins (SGBs). Metagenomic processing for the ZOE PREDICT1 cohort has previously been described[4] and taxonomic profiling was carried out using the same version of the reference database for MetaPhlAn (v.4.beta2) as for TwinsUK. Bray-Curtis dissimilarities were calculated using species-level relative abundances with the R package vegan (v.2.6.4), followed by principal coordinate analysis (PCoA) to visualise community variation. Ecologically abnormal samples were identified through density-based spatial clustering of the first two principal coordinates, implemented using the R package dbscan (v.1.2.0).

From the TwinsUK study population, 726 participants were included with metagenome data and 837 from the ZOE PREDICT1 cohort. In the TwinsUK cohort 2233 species were profiled (567 species present in >10% samples) and 1738 in the subset of the ZOE PREDICT1 cohort included (524 species present in >10% samples). Of the species present in >10% samples in each cohort, 472 species were shared across both cohorts. Individual microbiome community structure (α-diversity) was characterised using the Shannon index, calculated with the R package microbiome (v.1.24.0) on species-level data. Intra-individual microbiome diversity (β-diversity) was assessed using Bray-Curtis dissimilarities. The relative abundances of taxa were transformed using the centred log-ratio method prior to statistical analyses.

## Metabolomics profiling, compound identification and quantification

Untargeted metabolomics profiling was conducted using ultra high-performance liquid chromatography-tandem MS (UHPLC-MS/MS) by Metabolon Inc. (Morrisville, NC) as previously described in faeces[85] and serum[100] for TwinsUK. Sample preparation was conducted using the automated MicroLab STAR® system (Hamilton Company) and recovery standards were added during extraction for quality control. Macromolecules were precipitated using methanol, shaken for 2 min (Glen Mills GenoGrinder 2000), and then centrifuged. The extract was divided into aliquots for four UHPLC-MS/MS analyses: two reverse-phase (RP) methods using positive electrospray ionisation (ESI), one RP method with negative ESI, and one hydrophilic interaction chromatography (HILIC) method with negative ESI. Organic solvents were evaporated under warm Nitrogen (Argonaut SPE dry, Biotage), and if required, extracts were stored at −80 °C prior to analysis.

Controls included pooled matrix samples (technical replicates), extracted water blanks (process controls), and in-house reference material (human plasma). Additionally, every sample was spiked with internal QC standards (recovery and internal standards). Instrument variability (spiked standards) and overall process variability (endogenous metabolites in pooled samples) was assessed by median relative standard deviation (RSD). Samples and controls were randomised across the run. Metabolomic profiling was performed using a Waters ACQUITY UPLC system coupled to a Thermo Scientific Q-Exactive Orbitrap mass spectrometer with a heated electrospray ionisation (HESI-II) source, operated at 35,000 resolution. Reconstituted extracts were analysed by four complementary methods[101,102]. For RP separations, extracts were gradient-eluted (flow rate 350 μL/min) on a Waters BEH C18 column (2.1 × 100 mm, 1.7 μm particle size, 40 °C). Acidic extracts used mobile phases of (A) 0.1% formic acid in water and (B)

0.1% formic acid in methanol, with a gradient from 0% B to 70% B over 4 min, 70–98% B over 0.5 min, followed by 0.9 min at 98% B. Basic extracts were separated using (A) 6.5 mM ammonium bicarbonate in water (pH 8.0) and (B) 6.5 mM ammonium bicarbonate in 95:5 methanol/water, with the same gradient profile and flow rate. For polar metabolites, HILIC separations employed a Waters BEH Amide column under negative ionisation with appropriate mobile phases. Injection volume was 5 μL (with 2× overfill).

The Q-Exactive Orbitrap mass spectrometer was operated with a capillary temperature of 350 °C, sheath gas flow 40 a.u., aux gas flow 5 a.u., and spray voltage of 4.5 kV (positive mode) or 3.75 kV (negative mode). The mass spectrometer scanned from 70–1000 m/z, alternating between full MS and data-dependent MS/MS acquisition. MS/MS spectra were collected using higher-energy collisional dissociation (HCD, normalised collision energy 35–40 eV) with a 3 m/z isolation window, dynamic exclusion of 3.5 s, and approximately six scans per second (three MS and three MS/MS). For full detailed methods please see Evans et al.[103].

The metabolomic data sets measured by Metabolon included 526 (faeces) and 774 (serum) known metabolites belonging to the following broad categories: amino acids, peptides, carbohydrates, energy intermediates, lipids, nucleotides, cofactors and vitamins, and xeno-biotics. Metabolon's informatics pipeline comprised LIMS, proprietary peak-identification software, and visualisation tools. Raw spectral data were processed and deconvoluted using Metabolon-developed applications. Compounds were identified by matching retention indices (RI), accurate mass (±10 ppm), and MS/MS fragmentation spectra against a proprietary library (>3300 purified standards plus recurrent unnamed features). The RI is a normalised measure of chromatographic retention, calculated from a compound's elution time relative to surrounding retention markers, correcting for retention time drift across runs[103]. Compounds were quantified by calculating the AUC. Metabolite intensities were run day normalised ("block correction") in which compound medians were adjusted to 1.0 to correct for day-to-day instrument variability, followed by an inverse normal transformation to approximate a normal distribution. For data imputation, metabolites with <20% missing values were imputed to the observed minimum value for that metabolite. Metabolites with >20% missing values across samples were excluded from further analysis.

## Statistical analysis

Statistical analyses were conducted using R (v.4.3.2). The covariates were scaled to have a mean of 0 and a standard deviation of 1. Sex was coded as a factor variable: 0 for females or 1 for males.

**Random forest (RF) machine learning models.** For the prediction of adherence to dietary patterns and habitual intakes of food and beverage groups, the RF machine learning algorithm was used. RF models were chosen as they have been shown to effectively handle high dimensional metabolomics datasets[104,105]. RF models were constructed using the ranger package (v.0.16.0). Regression models predicted adherence to dietary patterns and habitual intake of food and beverage groups, whereas binary classification models were used to predict the top and bottom quartiles for each. For both types of models, the TwinsUK dataset was partitioned into train (80%) and test (20%) sets, ensuring that twin pairs remained in the same subset. For binary classifiers an equal class balance was maintained.

Feature selection was carried out using the Boruta algorithm with the R package, Boruta (v.8.0.0). The Boruta algorithm is a non-parametric, permutation-based algorithm for feature selection and was chosen as it has been shown as a stable feature selection approach for high-dimensional -omics datasets[106]. The training dataset was used for feature selection, and consisted of 650 faecal metabolites as well as the covariates, age, sex, and BMI. The Boruta algorithm was run separately, based on 500 iterations, for regression and classification models for

each index and food or beverage group. Features confirmed as important were then used as the input features for the subsequent models (Supplementary Data 3). For each dietary index or food group, two RF classification and two regression models were defined: (i) The null model in which only the covariates were used as input variables; and (ii) the metabolite model which included the covariates as well as the metabolites identified using the Boruta algorithm. For 10-year ASCVD risk prediction, to prevent data leakage, only BMI was included as a covariate due to age and sex being factors used to compute the score. Models included: (i) DASH score and BMI, (ii) faecal metabolites and BMI, (iii) faecal metabolites, DASH score and BMI.

Hyperparameter optimisation was carried out using the caret R package (v.6.0.94), applying five-fold cross validation repeated 10 times. A grid search of varying hyperparameter values was employed, including the number of variables randomly sampled at each split (mtry = square root of the number of input features, then 10%, 20%, 30%, 40% and 50% the number of input features), node size (5, 10, 15) and split rule (Binary Classifier: [gini, extratrees, hellinger]; Regression: [variance, extratrees, maxstat]) (Supplementary Data 1 and 2). The optimal model was selected based on root mean squared error (RMSE) for regression models or accuracy for binary classifiers.

Predictive power was assessed using the 20% testing set and the ZOE PREDICT1 cohort as an additional validation dataset. For binary classifiers, prediction class probabilities were used to define the ROC curve and AUC (R package pROC (v.1.18.5)). DeLong's test was used to determine if the AUC scores of the metabolite models were significantly different to their respective null models[107]. To confirm consistent performance and ensure broader applicability, we applied a stringent criterion by requiring that the AUCs obtained in the TwinsUK hold-out test set and ZOE PREDICT1 cohort were not significantly different by using a bootstrap approach. For each dataset (TwinsUK and ZOE PREDICT1), we generated 1000 bootstrap samples with replacement and computed the AUC for each resample. At each iteration, we calculated the difference in AUCs between cohorts and derived an empirical distribution of AUC differences. A two-sided $p$ value was then computed as the proportion of bootstrap iterations in which the absolute difference in AUC exceeded or equalled the observed difference. For regression models, the Spearman rank correlation coefficient was computed between the predicted values and the ground truth labels. The distribution of model predictions was again calculated using the same bootstrap with replacement approach.

To find the smallest number of input features with the greatest predictive power for each optimised metabolite model, an RFE approach was employed. Feature importance was computed using ranger's permutation variable importance approach[108], and then starting with the least important, each input feature was set to the mean value from the training dataset for that feature and the AUC score or Spearman rho coefficient was computed for the testing set and the ZOE PREDICT1 dataset. A trade-off threshold between number of input features and predictive power was assessed manually for each model. The smallest number of input features was selected that maintained an AUC score no less than 0.05 of the original AUC score in the TwinsUK hold-out test set and ZOE PREDICT1 validation set. The selected subsets of input features were included as part of predictive faecal metabolite panels for dietary index and food group models.

For the comparison of serum metabolites and faecal metabolites for predicting high and low adherence to dietary indices or reported intakes of robustly predicted food and beverage groups, the same machine learning framework was applied using RF binary classification models. A subset of TwinsUK ($n = 1618$) was used for both faecal and serum metabolite models. Predictive power of the models was assessed using the TwinsUK 20% hold-out test set.

**Univariate analysis.** Due to the skewed nature of the FFQ dietary data, the data were inverse-rank transformed prior to univariate analysis[109].

Associations between the dietary food groups, defined for the PDI index, and characterised faecal metabolites were assessed through linear mixed-effects regression models (LMER), fitted using restricted maximum likelihood (REML) via the lme4 R package (v.1.1-33) and tested with lmerTest (v.3.1.3). Only characterised metabolites were included due to their relevance as potential markers of food group intake. The models included age, BMI (at the time of faecal sample), and sex as covariates and were corrected for the non-independence of twin observations by including twin pair family structure as a random effect. For associations between the gut microbiome (α-diversity or species abundance) with food groups or faecal metabolites, LMER models were constructed in the same way. Results were adjusted for multiple testing using the Bonferroni correction. Due to the presence of highly correlated variables, the Bonferroni threshold was computed based on the effective number of independent tests ($M_{eff}$), determined by the eigenvalue variance of the correlation matrix[110].

Results from the LMER models with the same effect size direction in the discovery and replication cohorts were then meta-analysed using fixed-effects meta-analysis implemented via the R package metafor (v.4.4–0). Reported effect size and $p$ values from the meta-analysis were considered replicated across both cohorts if $p$ values from the meta-analysis were below the Bonferroni thresholds defined within the TwinsUK discovery cohort.

For associations between food and beverage groups and gut microbial species, LMER models were constructed and results combined in the same way as previously described. The resulting $p$ values from the fixed-effects meta-analysis were corrected for multiple testing using the Benjamini-Hochberg false discovery rate (FDR), with results considered significant.

Faecal or serum metabolites identified as important predictors for the RF binary classification models were assessed for their associations with gut microbial species in a subset of TwinsUK participants with gut metagenomics, faecal metabolomics and serum metabolomics data ($n = 657$). LMER models, adjusted for age, sex, BMI, and twin family structure were used. Association strength was quantified using the $-\log_{10}$ of the $p$ values obtained and the Wilcoxon rank sum test was used to compare association strengths between gut microbial species and either faecal or serum metabolites.

**Multivariate analysis.** The variance between intra-individual microbiome community structure, as explained by dietary-associated faecal metabolites, was determined by permutational analysis of variance (PERMANOVA) implemented via the R package vegan (v.2.6.4). Briefly, a distance matrix defined using the Bray-Curtis dissimilarity metric (β-diversity) was computed with species level relative abundance data. A PERMANOVA analysis based on 10,000 permutations was conducted separately for the TwinsUK and ZOE PREDICT1 cohorts for each of the dietary-associated faecal metabolites. The input formula for adonis2 was constructed in the following order to include the covariates age, sex, BMI, and finally the metabolite of interest. To account for the twin family structure present in both datasets, one twin from each family pair was randomly removed, resulting in a reduced sample size for both cohorts (TwinsUK = 474; ZOE PREDICT1 = 219). The permutation-based $p$ values were corrected for multiple testing by means of the Bonferroni correction. The coefficient of determination ($R^2$) was used to determine the percentage of variance attributed to the metabolite. The weighted combined $R^2$ values for both cohorts were calculated as the weighted average based on cohort sample sizes.

**Faecal metabolite pathway enrichment analysis.** Food group sets of significantly associated faecal metabolites were subject to ORA. For both positively and negatively associated metabolite sets, the probability of the observed overlap between the associated metabolites within the same KEGG pathway or subpathway and the total set of measured metabolites was determined using Fisher's exact test based

on the hypergeometric distribution[111]. The resulting *p* values were adjusted for multiple testing by means of the Bonferroni correction.

**Hierarchical clustering analysis.** Identified associations were visualised using ComplexHeatmap (v.2.18.0) and hierarchical clustering was performed using the Adjusted Coefficient of Commonality (ACC) as the distance measure. The ACC, first described by Posma et al.[20], is a novel measure of similarity between charged binary sets (A, B) and is particularly relevant for metabolomics when the direction of the association is important to consider[20]. Hierarchical clustering was applied to the identified metabolite and dietary associations using Ward's linkage method. The optimal number of clusters was determined by the maximum silhouette score generated for a range of cluster values.

**Mediation analysis.** Causal mediation analysis was performed using the mediation R package (v.4.5.0) to understand both:
a) The mediating effect of faecal metabolites between diet and microbial species.
b) The mediating effect of microbial species between diet and faecal metabolites.

LMER models for food groups, species and metabolites corrected for age, sex, BMI, and twin family structure were used as the input models. Mediatory effects were considered significant at $p < 0.05$.

**Reporting summary**
Further information on research design is available in the Nature Portfolio Reporting Summary linked to this article.

## Data availability
The raw metagenomic sequence data used in this study have been deposited in the European Bioinformatics Institute European Nucleotide Archive database (TwinsUK accession code: PRJEB98467; ZOE PREDICT1 accession code: PRJEB39223). All data relating to TwinsUK samples have been deposited to the TwinsUK BioResource data management team. These data and non-metagenomic data for ZOE PREDICT1 are available by application to the Twin Research Executive Access committee (TREC) at King's College London. The TwinsUK BioResource is managed by TREC, which provides governance of access to TwinsUK data and samples. TwinsUK data users are bound by data sharing agreement set out in the data access application form, which includes responsibilities with respect to third party data sharing and maintaining participant privacy. Further responsibilities include a responsibility to acknowledge data sharing. All results from the associations studies between food and beverage groups, faecal metabolites or gut microbial species can be accessed at https://twinsuk.ac.uk/publisheddata_gutexplorer/.

## Code availability
Metagenomic data analysis was carried out using the YAMP pipeline (v.0.9.5.0)[98] (https://github.com/alesssia/YAMP), with taxonomic profiling and species level microbial relative abundances computed using MetaPhlAn (v.4.beta.2)[99] (https://github.com/biobakery/MetaPhlAn).

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

## Acknowledgements

We thank all the volunteers from the TwinsUK and ZOE PREDICT1 studies for contributing and supporting this research. M.F. and R.P. are funded by the Chronic Disease Research Foundation (CDRF) grant 27/2023. C.M. is funded by the CDRF, the Italian Ministry of Education and Research (MUR): Dipartimenti di Eccellenza Program 2023–2027 and by the Italian Ministry of Health—Bando Ricerca Corrente. TwinsUK is funded by the Wellcome Trust, Medical Research Council, Versus Arthritis, European Union Horizon 2020, CDRF, Wellcome Leap Dynamic Resilience Programme (co-funded by Temasek Trust), Zoe Ltd, the National Institute for Health and Care Research (NIHR) Clinical Research Network (CRN) and Biomedical Research Centre based at Guy's and St Thomas' NHS Foundation Trust in partnership with King's College London. Data analysis was carried out using the King's Computational Research, Engineering and Technology Environment (CREATE)[112]. Schematics were created with Biorender.com.

## Author contributions

R.P., M.F. and C.M. had complete access to the data in the study. A.V. processed and completed the taxonomic profiling of the TwinsUK shotgun metagenomics data. R.P. and M.F. designed the study. R.P. performed the data analysis and visualisation. R.P., M.F. and C.M. drafted the manuscript. R.P., M.F., C.M., A.V., X.Z., P.L., A.F.B., Y.L., F.A., K.B., K.E.W., G.A.M., J.W., N.S., S.E.B., T.D.S., E.R.L. and R.G. contributed to the acquisition of the data, analysis and interpretation of results as well as revision of the manuscript.

## Competing interests

J.W. and T.D.S. are cofounders of ZOE. N.S., F.A., S.E.B. and T.D.S. are consultants to ZOE. J.W., K.M.B. and E.R.L. are or have been employees of ZOE. T.D.S., J.W., N.S., F.A. and S.E.B. receive options with ZOE. K.E.W. and G.A.M. are employees of Metabolon, Inc. The remaining authors declare no competing interests.
