## [Transparent Peer Review file · Nature Communications]

Faecal metabolites as a readout of habitual diet capture dietary interactions with the gut microbiome

Corresponding Author: Dr Mario Falchi

Version 0:

Reviewer comments:

Reviewer #1

(Remarks to the Author)

The authors have addressed those raised concerns. There are no further comments.

Reviewer #2

(Remarks to the Author)

The authors have addressed all my concerns. My final remark is related to the Supplementary table 6 that was added to provide the analytical data for the unknown metabolites: What does RI stand for and what are the units, which analytical mode was the compound detected at (column and ionization for the m/z value), is there ms/ms fragmentation data for the compound? After these final modifications, I am happy to recommend the manuscript for publication.

Reviewer #3

(Remarks to the Author)

The authors answered all my comments, I have only few minor points:

Please make sure that you clearly state in the method information on time differences of FFQ collection and metabolomics measurements, and that metabolomics imputation method and used thresholds are also incorporated into the methods section.

Response to Reviewers

Reviewer #1 (Remarks to the Author):

The authors have addressed those raised concerns. There are no further comments.

Author Response: We are pleased to have addressed the concerns of the reviewer.

Reviewer #2 (Remarks to the Author):

The authors have addressed all my concerns. My final remark is related to the Supplementary table 6 that was added to provide the analytical data for the unknown metabolites: What does RI stand for and what are the units, which analytical mode was the compound detected at (column and ionization for the m/z value), is there ms/ms fragmentation data for the compound? After these final modifications, I am happy to recommend the manuscript for publication.

Author Response: In Supplementary Table 6, RI stands for Retention Index. The RI derived by Metabolon is a normalised measure of chromatographic retention, calculated from a compound's elution time relative to surrounding retention markers, correcting for retention time drift across runs, essentially the retention time multiplied by 1000. For example, something at 1000 RI is 1 minute. The definition of the RI has been added to the methods as well as the description of Supplementary Table 6. We have further removed the abbreviation RI in Supplementary Table 6 and included the term "Retention Index".

Regarding the analytical mode and MS/MS fragmentation data, metabolites were measured using four complementary UHPLC-MS/MS methods, including two reverse-phase (RP) separations in positive electrospray ionization (ESI), one RP separation in negative ESI, and one hydrophilic interaction chromatography (HILIC) separation in negative ESI. All analyses were performed on a Waters ACQUITY UPLC system coupled to a Thermo Scientific Q-Exactive Orbitrap mass spectrometer, with data-dependent MS/MS acquisition for structural confirmation of metabolites. Higher-energy collisional dissociation (HCD) MS/MS spectra were collected for all compounds. Metabolon began releasing spectral data in 2024 (<https://www.metabolon.com/news/spectral-data-files-global-discovery-projects/>). Although we missed the initial window, Metabolon has agreed to provide global spectral data for 200 samples free of charge.

Access will be managed by the TwinsUK Executive Committee for bona fide researchers (see Data Availability).

Reviewer #3 (Remarks to the Author):

The authors answered all my comments, I have only few minor points: Please make sure that you clearly state in the method information on time differences of FFQ collection and metabolomics measurements, and that metabolomics imputation method and used thresholds are also incorporated into the methods section.

Author Response: We are pleased to have addressed all the comments from the Reviewer. The time differences between FFQ and metabolomics measurements have been described in the methods of the paper:

“Here we included 1,810 individuals with faecal metabolomics profiling (Metabolon inc) who completed an adapted 131-item European Prospective Investigation into Cancer and Nutrition (EPIC) Food Frequency Questionnaire (FFQ)⁸⁴ within 3 years (mean time = 1.4 years (± 1)) of their faecal sample collection.”

We have updated the methods to directly specify the imputation method for the metabolomics data, including the thresholds used:

“For data imputation, metabolites with <20% missing values were imputed to the observed minimum value for that metabolite. Metabolites with >20% missing values across samples were excluded from further analysis.”